# Hamiltonian Monte Carlo using an adjoint-differentiated Laplace approximation: Bayesian inference for latent Gaussian models and beyond

**Charles C. Margossian**
Department of Statistics
Columbia University
New York, NY 10027
charles.margossian@columbia.edu

**Aki Vehtari**
Department of Computer Science
Aalto University
02150 Espoo, Finland
Finnish Center for Artificial Intelligence

**Daniel Simpson**
Department of Statistical Sciences
University of Toronto
ON M5S, Canada

**Raj Agrawal**
CSAIL
Massachusetts Institute of Technology
Cambridge, MA 02139

## Abstract

Gaussian latent variable models are a key class of Bayesian hierarchical models with applications in many fields. Performing Bayesian inference on such models can be challenging as Markov chain Monte Carlo algorithms struggle with the geometry of the resulting posterior distribution and can be prohibitively slow. An alternative is to use a Laplace approximation to marginalize out the latent Gaussian variables and then integrate out the remaining hyperparameters using dynamic Hamiltonian Monte Carlo, a gradient-based Markov chain Monte Carlo sampler. To implement this scheme efficiently, we derive a novel adjoint method that propagates the minimal information needed to construct the gradient of the approximate marginal likelihood. This strategy yields a scalable differentiation method that is orders of magnitude faster than state of the art differentiation techniques when the hyperparameters are high dimensional. We prototype the method in the probabilistic programming framework Stan and test the utility of the embedded Laplace approximation on several models, including one where the dimension of the hyperparameter is ∼6,000. Depending on the cases, the benefits can include an alleviation of the geometric pathologies that frustrate Hamiltonian Monte Carlo and a dramatic speed-up.

## 1 Introduction

Latent Gaussian models observe the following hierarchical structure:

$$\phi \sim \pi(\phi), \qquad \theta \sim \mathrm{Normal}(0, K(\phi)), \qquad y \sim \pi(y \mid \theta, \phi).$$

Typically, single observations $y_i$ are independently distributed and only depend on a linear combination of the latent variables, that is $\pi(y_i \mid \theta, \phi) = \pi(y_i \mid a_i^T \theta, \phi)$, for some appropriately defined vectors $a_i$. This general framework finds a broad array of applications: Gaussian processes, spatial models, and multilevel regression models to name a few examples. We denote $\theta$ the *latent Gaussian variable* and $\phi$ the *hyperparameter*, although we note that in general $\phi$ may refer to any latent variable other than $\theta$. Note that there is no clear consensus in the literature on what constitutes a "latent Gaussian model"; we use the definition from the seminal work by Rue et al. [34].

We derive a method to perform Bayesian inference on latent Gaussian models, which scales when $\phi$ is high dimensional and can handle the case where $\pi(\phi \mid y)$ is multimodal, provided the energy barrier between the modes is not too strong. This scenario arises in, for example, general linear models with a regularized horseshoe prior [13] and in sparse kernel interaction models [1]. The main application for these models is studies with a low number of observations but a high-dimensional covariate, as seen in genomics.

The inference method we develop uses a gradient-based Markov chain Monte Carlo (MCMC) sampler, coupled with a Laplace approximation to marginalize out $\theta$. The key to successfully implementing this scheme is a novel adjoint method that efficiently differentiates the approximate marginal likelihood. In the case of a classic Gaussian process (Section 4), where $\dim(\phi) = 2$, the computation required to evaluate and differentiate the marginal is on par with the GPstuff package [40], which uses the popular algorithm by Rasmussen and Williams [33]. The adjoint method is however orders of magnitude faster when $\phi$ is high dimensional. Figure 1 shows the superior scalability of the adjoint method on simulated data from a sparse kernel interaction model. We lay out the details of the algorithms and the experiment in Section 3.

## 1.1 Existing methods

Bayesian computation is, broadly speaking, split between two approaches: (i) MCMC methods that approximately sample from the posterior, and (ii) approximation methods in which one finds a tractable distribution that approximates the posterior (e.g. variational inference, expectation propagation, and asymptotic approximations). The same holds for latent Gaussian models, where we can consider (i) Hamiltonian Monte Carlo (HMC) sampling [30, 5] and (ii) approximation schemes such as variational inference (VI) [10] or marginalizing out the latent Gaussian variables with a Laplace approximation before deterministically integrating the hyperparameters [38, 34].

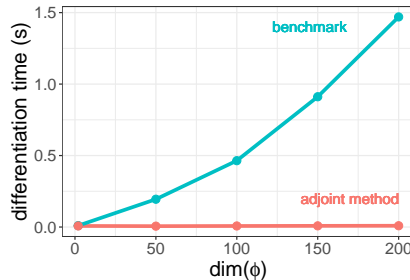

Figure 1: Wall time to differentiate the marginal density using the adjoint method (Algorithm 2) and, as a benchmark, the method by Rasmussen and Williams [33] (Algorithm 1).

**Hamiltonian Monte Carlo sampling.** When using MCMC sampling, the target distribution is

$$\pi(\theta, \phi \mid y) \propto \pi(y \mid \theta, \phi)\pi(\theta \mid \phi)\pi(\phi),$$

and the Markov chain explores the joint parameter space of $\theta$ and $\phi$.

HMC is a class of MCMC algorithms that powers many modern probabilistic programming languages, including Stan [12], PyMC3 [36], and TensorFlow Probability [16]. Its success is both empirically and theoretically motivated (e.g. [9]) and, amongst other things, lies in its ability to probe the geometry of the target distribution via the gradient. The algorithm is widely accessible through a combination of its dynamic variants [22, 5], which spare the users the cumbersome task of manually setting the algorithm's tuning parameters, and automatic differentiation, which alleviates the burden of calculating gradients by hand (e.g. [27, 2, 21]). There are known challenges when applying HMC to hierarchical models, because of the posterior distribution's problematic geometry [7]. In the case of latent Gaussian models, this geometric grief is often caused by the latent Gaussian variable, $\theta$, and its interaction with $\phi$. Certain samplers, such as Riemannian HMC [19, 4] and semi-separable HMC [44], are designed to better handle difficult geometries. While promising, these methods are difficult to implement, computationally expensive, and to our knowledge not widely used.

**Variational inference.** VI proposes to approximate the target distribution, $\pi(\theta, \phi \mid y)$, with a tractable distribution, $q(\theta, \phi)$, which minimizes the Kullback-Leibler divergence between the approximation and the target. The optimization is performed over a pre-defined family of distributions, $\mathcal{Q}$. Adaptive versions, such as black-box VI [32] and automatic differentiation VI (ADVI) [25], make it easy to run the algorithm. VI is further made accessible by popular software libraries, including the above-mentioned probabilistic programming languages, and others packages such as GPyTorch for Gaussian processes [18]. For certain problems, VI is more scalable than MCMC, because it can

be computationally much faster to solve an optimization problem than to generate a large number of samples. There are however known limitations with VI (e.g. [10, 43, 23, 15]). Of interest here is that $\mathcal{Q}$ may not include appropriate approximations of the target: mean field or full rank Gaussian families, for instance, will underestimate variance and settle on a single mode, even if the posterior is multimodal (e.g. [43]).

**Marginalization using a Laplace approximation.** The embedded Laplace approximation is a popular algorithm, and a key component of the R packages INLA (*integrated nested Laplace integration*, [34, 35]) and TMB (*template model builder*, [24]), and the GPstuff package [40]. The idea is to marginalize out $\theta$ and then use standard inference techniques on $\phi$.

We perform the Laplace approximation

$$\pi(\theta \mid \phi, y) \approx \pi_{\mathcal{G}}(\theta \mid y, \phi) := \text{Normal}(\theta^*, \Sigma^*),$$

where $\theta^*$ matches the mode and $[\Sigma^*]^{-1}$ the curvature of $\pi(\theta \mid \phi, y)$. Then, the marginal posterior distribution is approximated as follows:

$$\pi(\phi \mid y) \approx \pi_{\mathcal{G}}(\phi \mid y) := \pi(\phi) \frac{\pi(\theta^* \mid \phi)\pi(y \mid \theta^*, \phi)}{\pi_{\mathcal{G}}(\theta^* \mid \phi, y)\pi(y)}.$$

Once we perform inference on $\phi$, we can recover $\theta$ using the conditional distribution $\pi_{\mathcal{G}}(\theta \mid \phi, y)$ and effectively marginalizing $\phi$ out. For certain models, this yields much faster inference than MCMC, while retaining comparable accuracy [34]. Furthermore the Laplace approximation as a marginalization scheme enjoys very good theoretical properties [38].

In the R package INLA, approximate inference is performed on $\phi$, by characterizing $\pi(\phi \mid y)$ around its presumed mode. This works well for many cases but presents two limitations: the posterior must be well characterized in the neighborhood of the estimated mode and it must be low dimensional, *"2–5, not more than 20"* [35]. In one of the examples we study, the posterior of $\phi$ is both high dimensional ($\sim$6000) and multimodal.

**Hybrid methods.** Naturally we can use a more flexible inference method on $\phi$ such as a standard MCMC, as discussed by Gómez-Rubio and Rue [20], and HMC as proposed in GPstuff and TMB, the latter through its extension TMBStan and AdNuts (*automatic differentiation with a No-U-Turn Sampler* [28]). The target distribution of the HMC sampler is now $\pi_{\mathcal{G}}(\phi \mid y)$.

To use HMC, we require the gradient of $\log \pi_{\mathcal{G}}(y \mid \phi)$ with respect to $\phi$. Much care must be taken to ensure an efficient computation of this gradient. TMB and GPstuff exemplify two approaches to differentiate the approximate marginal density. The first uses automatic differentiation and the second adapts the algorithms in Rasmussen and Williams [33]. One of the main bottlenecks is differentiating the estimated mode, $\theta^*$. In theory, it is straightforward to apply automatic differentiation, by brute-force propagating derivatives through $\theta^*$, that is, sequentially differentiating the iterations of a numerical optimizer. But this approach, termed the *direct method*, is prohibitively expensive. A much faster alternative is to use the implicit function theorem (e.g. [3, 27]). Given any accurate numerical solver, we can always use the implicit function theorem to get derivatives, as notably done in the Stan Math Library [11] and in TMB's *inverse subset algorithm* [24]. One side effect is that the numerical optimizer is treated as a black box. By contrast, Rasmussen and Williams [33] define a bespoke Newton method to compute $\theta^*$, meaning we can store relevant variables from the final Newton step when computing derivatives. In our experience, this leads to important computational savings. But overall this method is much less flexible, working well only when $\phi$ is low dimensional and requiring the user to pass the tensor of derivatives, $\partial K / \partial \phi$.

## 2 Aim and results of the paper

We improve the computation of HMC with an embedded Laplace approximation. Our implementation accommodates any covariance matrix $K$, without requiring the user to specify $\partial K / \partial \phi$, efficiently differentiates $\log \pi_{\mathcal{G}}(y \mid \phi)$, even when $\phi$ is high dimensional, and deploys dynamic HMC to perform inference on $\phi$. We introduce a novel adjoint method to differentiate $\log \pi_{\mathcal{G}}(y \mid \phi)$, build the algorithm in C++, and add it to the Stan language. Our approach combines the Newton solver of Rasmussen and Williams [33] with a non-trivial application of automatic differentiation.

Equipped with this implementation, we test dynamic HMC with an embedded Laplace approximation on a range of models, including ones with a high dimensional and multimodal hyperparameter. We do so by benchmarking our implementation against Stan's dynamic HMC, which runs MCMC on both the hyperparameter and the latent Gaussian variable. For the rest of the paper, we call this standard use of dynamic HMC, *full HMC*. We refer to marginalizing out $\theta$ and using dynamic HMC on $\phi$, as the *embedded Laplace approximation*. Another candidate benchmark is Stan's ADVI. Yao et al. [43] however report that ADVI underestimates the posterior variance and returns a unimodal approximation, even when the posterior is multimodal. We observe a similar behavior in the models we examine. For clarity, we relegate most of our analysis on ADVI to the Supplementary Material.

Our computer experiments identify cases where the benefits of the embedded Laplace approximation, as tested with our implementation, are substantial. In the case of a classic Gaussian process, with $\dim(\phi) = 2$ and $\dim(\theta) = 100$, we observe an important computational speed up, when compared to full HMC. We next study a general linear regression with a sparsity inducing prior; this time $\dim(\phi) \approx 6,000$ and $\dim(\theta) \approx 100$. Full HMC struggles with the posterior's geometry, as indicated by divergent transitions, and requires a model reparameterization and extensive tuning of the sampler. On the other hand, the embedded Laplace approximation evades many of the geometric problems and solves the approximate problem efficiently. We observe similar results for a sparse kernel interaction model, which looks at second-order interactions between covariates [1]. Our results stand in contrast to the experiments presented in Monnahan and Kristensen [28], who used a different method to automatically differentiate the Laplace approximation and reported at best a minor speed up. We do however note that the authors investigated different models than the ones we study here.

In all the studied cases, the likelihood is log-concave. Combined with a Gaussian prior, log-concavity guarantees that $\pi(\theta \mid \phi, y)$ is unimodal. Detailed analysis on the error introduced by the Laplace approximation for log-concave likelihoods can be found in references (e.g. [26, 39, 14, 41]) and are consistent with the results from our computer experiments.

## 3  Implementation for probabilistic programming

In order to run HMC, we need a function that returns the approximate log density of the marginal likelihood, $\log \pi_{\mathcal{G}}(y \mid \phi)$, and its gradient with respect to $\phi$, $\nabla_\phi \log \pi_{\mathcal{G}}(y \mid \phi)$. The user specifies the observations, $y$, and a function to generate the covariance $K$, based on input covariates $x$ and the hyperparameters $\phi$. In the current prototype, the user picks the likelihood, $\pi(y \mid \theta, \phi)$, from a set of options[1]: for example, a likelihood arising from a Bernoulli distribution with a logit link.

Standard implementations of the Laplace approximation use the algorithms in Rasmussen and Williams [33, chapter 3 and 5] to compute (i) the mode $\theta^*$ and $\log \pi_{\mathcal{G}}(y \mid \phi)$, using a Newton solver; (ii) the gradient $\nabla_\phi \log \pi_{\mathcal{G}}(y \mid \phi)$ (Algorithm 1), and (iii) simulations from $\pi_{\mathcal{G}}(\theta \mid y, \phi)$. The major contribution of this paper is to construct a new differentiation algorithm, i.e. item (ii).

### 3.1  Using automatic differentiation in the algorithm of Rasmussen and Williams [33]

The main difficulty with Algorithm 1 from Rasmussen and Williams [33] is the requirement for $\partial K / \partial \phi_j$ at line 8. For classic problems, where $K$ is, for instance, an exponentiated quadratic kernel, the derivatives are available analytically. This is not the case in general and, in line with the paradigm of probabilistic programming, we want a method that does not require the user to specify the tensor of derivatives, $\partial K / \partial \phi$.

Automatic differentiation allows us to numerically evaluate $\partial K / \partial \phi$ based on computer code to evaluate $K$. To do this, we introduce the map $\mathcal{K}$

$$\mathcal{K} \quad : \quad \mathbb{R}^p \to \mathbb{R}^{n(n+1)/2}$$
$$\phi \to K,$$

where $p$ is the dimension of $\phi$ and $n$ that of $\theta$. To obtain the full tensor of derivatives, we require either $p$ forward mode sweeps or $n(n+1)/2$ reverse mode sweeps. Given the scaling, we favor forward mode and this works well when $p$ is small. However, once $p$ becomes large, this approach is spectacularly inefficient.

**Algorithm 1** *Gradient of the approximate marginal density, $\pi_{\mathcal{G}}(y \mid \phi)$, with respect to the hyperparameters $\phi$, adapted from algorithm 5.1 by Rasmussen and Williams [33, chapter 5]. We store and reuse terms computed during the final Newton step, algorithm 3.1 in Rasmussen and Williams [33, chapter 3].*

---

    **input:** $y$, $\phi$, $\pi(y \mid \theta, \phi)$
2: **saved input from the Newton solver**: $\theta^*$, $K$, $W^{\frac{1}{2}}$, $L$, $a$
    $Z = \frac{1}{2}a^T\theta^* + \log\pi(y \mid \theta^*, \phi) - \sum \log(\mathrm{diag}(L))$
4: $R = W^{\frac{1}{2}}L^T \setminus (L \setminus W^{\frac{1}{2}})$
    $C = L \setminus (W^{\frac{1}{2}}K)$
6: $s_2 = -\frac{1}{2}\mathrm{diag}(\mathrm{diag}(K) - \mathrm{diag}(C^TC))\nabla_\theta^3 \log\pi(y \mid \theta^*, \phi)$
    **for** $j = 1 \dots \dim(\phi)$
8:     $K' = \partial K/\partial\phi_j$
        $s_1 = \frac{1}{2}a^TK'a - \frac{1}{2}\mathrm{tr}(RK')$
10:     $b = K'\nabla_\theta \log\pi(y \mid \theta, \phi)$
        $s_3 = b - KRb$
12:     $\frac{\partial}{\partial\phi_j}\pi(y \mid \phi) = s_1 + s_2^Ts_3$
    **end for**
14: **return** $\nabla_\phi \log\pi_{\mathcal{G}}(y \mid \phi)$

---

### 3.2 Adjoint method to differentiate the approximate log marginal density

To evaluate the gradient of a composite map, it is actually not necessary to compute the full Jacobian matrix of intermediate operations. This is an important, if often overlooked, property of automatic differentiation and the driving principle behind *adjoint methods* (e.g. [17]). This idea motivates an algorithm that does not explicitly construct $\partial K/\partial\phi$, a calculation that is both expensive and superfluous. Indeed, it suffices to evaluate $w^T\partial K/\partial\phi$ for the correct *cotangent vector*, $w^T$, an operation we can do in a single reverse mode sweep of automatic differentiation.

**Theorem 1** *Let $\log\pi_{\mathcal{G}}(y \mid \phi)$ be the approximate log marginal density in the context of a latent Gaussian model. Let $a$ be defined as in the Newton solver by Rasmussen and Williams [33, chapter 3], and let $R$ and $s_2$ be defined as in Algorithm 1. Then*

$$\nabla_\phi \log\pi_{\mathcal{G}}(y \mid \phi) = w^T\frac{\partial K}{\partial\phi},$$

*where the gradient is with respect to $\phi$ and*

$$w^T = \frac{1}{2}aa^T - \frac{1}{2}R + (s_2 + RKs_2)[\nabla_\theta \log\pi(y \mid \theta, \phi)]^T.$$

The proof follows from Algorithm 1 and noting that all the operations in $\partial K/\partial\phi_j$ are linear. We provide the details in the Supplementary Material. Armed with this result, we build Algorithm 2, a method that combines the insights of Rasmussen and Williams [33] with the principles of adjoint methods.

---

**Algorithm 2** *Gradient of the approximate marginal log density, $\log\pi_{\mathcal{G}}(y \mid \phi)$, with respect to the hyperparameters, $\phi$, using reverse mode automatic differentiation and theorem 1.*

---

    **input:** $y$, $\phi$, $\pi(y \mid \theta, \phi)$
2: Do lines 2 - 6 of Algorithm 1.
    Initiate an expression graph for automatic differentiation with $\phi_v = \phi$.
4: $K_v = \mathcal{K}(\phi_v)$
    $w^T = \frac{1}{2}aa^T - \frac{1}{2}R + (s_2 + RKs_2)[\nabla_\theta \log\pi(y \mid \theta, \phi)]^T$
6: Do a reverse sweep over $K$, with $w^T$ as the initial cotangent to obtain $\nabla_\phi \log\pi_{\mathcal{G}}(y \mid \phi)$.
    **return:** $\nabla_\phi \log\pi_{\mathcal{G}}(y \mid \phi)$.

---

Figure 1 shows the time required for one evaluation and differentiation of $\log\pi_{\mathcal{G}}(y \mid \phi)$ for the sparse kernel interaction model developed by Agrawal et al. [1] on simulated data. The covariance structure

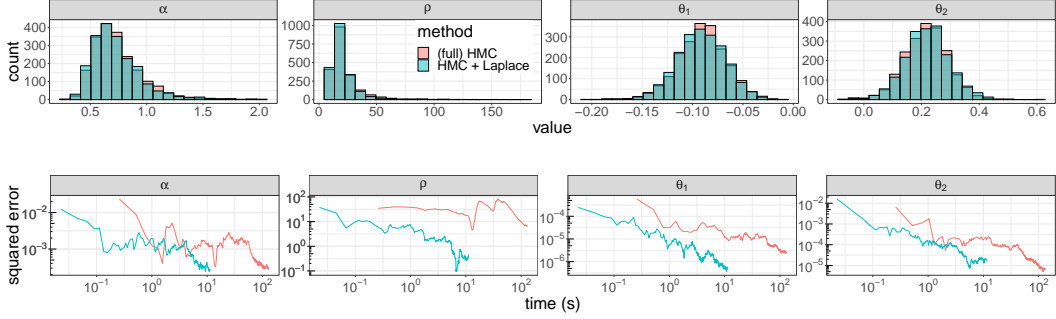

Figure 2: (Up) Posterior samples obtained with full HMC and the embedded Laplace approximation when fitting the disease map. (Down) Error when estimating the expectation value against wall time. Unreported in the figure is that we had to fit full HMC twice before obtaining good tuning parameters.

of this model is nontrivial and analytical derivatives are not easily available. We simulate a range of data sets for varying dimensions, $p$, of $\phi$. For low dimensions, the difference is small; however, for $p = 200$, Algorithm 2 is more than 100 times faster than Algorithm 1, requiring 0.009 s, instead of 1.47 s.

## 4 Gaussian process with a Poisson likelihood

We fit the disease map of Finland by Vanhatalo et al. [39] which models the mortality count across the country. The data is aggregated in $n = 911$ grid cells. We use 100 cells, which allows us to fit the model quickly both with full HMC and HMC using an embedded Laplace approximation. For the $i^{\text{th}}$ region, we have a 2-dimensional coordinate $x_i$, the counts of deaths $y_i$, and the standardized expected number of deaths, $y_e^i$. The full latent Gaussian model is

$$(\rho, \alpha) \sim \pi(\rho, \alpha), \qquad \theta \sim \text{Normal}(0, K(\alpha, \rho, x)), \qquad y_i \sim \text{Poisson}(y_e^i e^{\theta_i}),$$

where $K$ is an exponentiated quadratic kernel, $\alpha$ is the marginal standard deviation and $\rho$ the characteristic length scale. Hence $\phi = (\alpha, \rho)$.

Fitting this model with MCMC requires running the Markov chains over $\alpha$, $\rho$, and $\theta$. Because the data is sparse — one observation per group — the posterior has a funnel shape which can lead to biased MCMC estimates [29, 7]. A useful diagnostic for identifying posterior shapes that challenge the HMC sampler is *divergent transitions*, which occur when there is significant numerical error in the computation of the Markov chain trajectory [5].

To remedy these issues, we reparameterize the model and adjust the *target acceptance rate*, $\delta_a$. $\delta_a$ controls the precision of HMC, with the usual trade-off between accuracy and speed. For well behaved problems, the optimal value is 0.8 [8] but posteriors with highly varying curvature require a higher value. Moreover, multiple attempts at fitting the model must be done before we correctly tune the sampler and remove all the divergent transitions. See the Supplementary Material for more details.

An immediate benefit of the embedded Laplace approximation is that we marginalize out $\theta$ and only run HMC on $\alpha$ and $\rho$, a two-dimensional and typically well behaved parameter space. In the case of the disease map, we do not need to reparameterize the model, nor adjust $\delta_a$.

We fit the models with both methods, using 4 chains, each with 500 warmup and 500 sampling iterations. A look at the marginal distributions of $\alpha$, $\rho$, and the first two elements of $\theta$ suggests the posterior samples generated by full HMC and the embedded Laplace approximation are in close agreement (Figure 2). With a Poisson likelihood, the bias introduced by the Laplace approximation is small, as shown by Vanhatalo et al. [39]. We benchmark the Monte Carlo estimates of both methods against results from running 18,000 MCMC iterations. The embedded Laplace approximations yields comparable precision, when estimating expectation values, and is an order of magnitude faster (Figure 2). In addition, we do not need to tune the algorithm and the MCMC warmup time is much shorter ($\sim$10 seconds against $\sim$200 seconds for full HMC).

Table 1: Top six covariate indices, $i$, with the highest $90^{th}$ quantiles of $\log \lambda_i$ for the general linear model with a regularized horseshoe prior. The first two methods are in good agreement; ADVI selects different covariates, in part because it approximates the multimodal posterior with a unimodal distribution (see the Supplementary Material).

| (full) HMC | 2586 | 1816 | 4960 | 4238 | 4843 | 3381 |
|---|---|---|---|---|---|---|
| HMC + Laplace | 2586 | 1816 | 4960 | 4647 | 4238 | 3381 |
| ADVI | 1816 | 2416 | 4284 | 2586 | 5279 | 4940 |

## 5 General linear regression model with a regularized horseshoe prior

Consider a regression model with $n$ observations and $p$ covariates. In the "$p \gg n$" regime, we typically need additional structure, such as sparsity, for accurate inference. The horseshoe prior [13] is a useful prior when it is assumed that only a small portion of the regression coefficients are non-zero. Here we use the regularized horseshoe prior by Piironen and Vehtari [31]. The horseshoe prior is parameterized by a global scale term, the scalar $\tau$, and local scale terms for each covariate, $\lambda_j, j = 1, \ldots, p$. Consequently the number of hyperparameters is $\mathcal{O}(p)$.

To use the embedded Laplace approximation, we recast the regularized linear regression as a latent Gaussian model. The benefit of the approximation is not a significant speedup, rather an improved posterior geometry, due to marginalizing $\theta$ out. This means we do not need to reparameterize the model, nor fine tune the sampler. To see this, we examine the genetic microarray classification data set on prostate cancer used by Piironen and Vehtari [31] and fit a regression model with a Bernoulli distribution and a logit link. Here, $\dim(\theta) = 102$ and $\dim(\phi) = 5,966$.

We use 1,000 iterations to warm up the sampler and 12,000 sampling iterations. Tail quantiles, such as the $90^{th}$ quantile, allow us to identify parameters which have a small local shrinkage and thence indicate relevant covariates. The large sample size is used to reduce the Monte Carlo error in our estimates of these extreme quantiles.

Fitting this model with full HMC requires a fair amount of work: the model must be reparameterized and the sampler carefully tuned, after multiple attempts at a fit. We use a non-centered parameterization, set $\delta_a = 0.999$ (after attempting $\delta_a = 0.8$ and $\delta_a = 0.99$) and do some additional adjustments. Even then we obtain 13 divergent transitions over 12,000 sampling iterations. The Supplementary Material describes the tuning process in all its thorny details. By contrast, running the embedded Laplace approximation with Stan's default tuning parameters produces 0 divergent transitions. Hence the approximate problem is efficiently solved by dynamic HMC. Running ADVI on this model is also straightforward.

Table 1 shows the covariates with the highest $90^{th}$ quantiles, which are softly selected by full HMC, the embedded Laplace approximation and ADVI. For clarity, we exclude ADVI from the remaining figures but note that it generates, for this particular problem, strongly biased inference; more details can be found in the Supplementary Material. Figure 3 compares the expected probability of

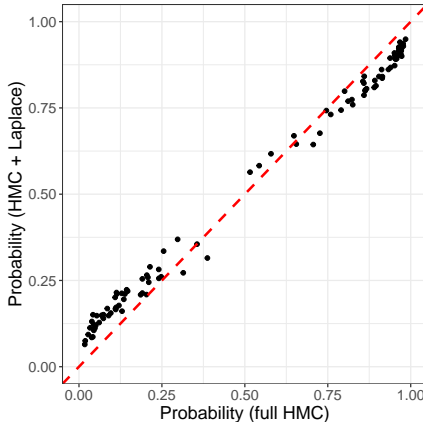

Figure 3: Expectation value for the probability of developing prostate cancer, as estimated by full HMC and HMC using an embedded Laplace approximation.

developing cancer. Figure 4 compares the posterior samples and the error when estimating various quantities of interest, namely (i) the expectation value of the global shrinkage, $\tau$, and the slab parameter, $c_{aux}$; and (ii) the $90^{th}$ quantile of two local shrinkage parameters. As a benchmark we use estimates obtained from 98,000 MCMC iterations.

The Laplace approximation yields slightly less extreme probabilities of developing cancer than the corresponding full model. This behavior is expected for latent Gaussian models with a Bernoulli

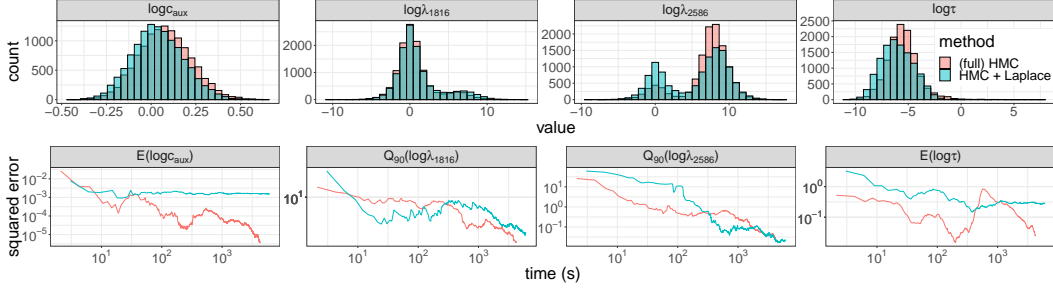

Figure 4: (Up) Posterior samples obtained with full HMC and HMC using an embedded Laplace approximation when fitting a general linear regression with a regularized horseshoe prior. (Down) Error when estimating various quantities of interest against wall time. $E$ stands for "expectation" and $Q_{90}$, "90$^{\text{th}}$ quantile". Unreported in the figure is that we had to run full HMC four times before obtaining reasonable tuning parameters.

Table 2: Top six covariate indices, $i$, with the highest 90$^{\text{th}}$ quantiles of $\log \lambda_i$ for the SKIM.

| **(full) HMC** | 2586 | 2660 | 2679 | 2581 | 2620 | 2651 |
|---|---|---|---|---|---|---|
| **HMC + Laplace** | 2586 | 2679 | 2660 | 2581 | 2620 | 2548 |
| **ADVI** | 2586 | 2526 | 2106 | 2550 | 2694 | 2166 |

observation model, and has been studied in the cases of Gaussian processes and Gaussian random Markov fields (e.g. [26, 14, 41]). While introducing a bias, the embedded Laplace approximation yields accuracy comparable to full HMC when evaluating quantities of interest.

## 6 Sparse kernel interaction model

A natural extension of the general linear model is to include interaction terms. To achieve better computational scalability, we can use the kernel interaction trick by Agrawal et al. [1] and build a sparse kernel interaction model (SKIM), which also uses the regularized horseshoe prior by Piironen and Vehtari [31]. The model is an explicit latent Gaussian model and uses a non-trivial covariance matrix. The full details of the model are given in the Supplementary Material.

When fitting the SKIM to the prostate cancer data, we encounter similar challenges as in the previous section: ~150 divergent transitions with full HMC when using Stan's default tuning parameters. The behavior when adding the embedded Laplace approximation is much better, although there are still ~3 divergent transitions,[2] which indicates that this problem remains quite difficult even after the approximate marginalization. We also find large differences in running time. The embedded Laplace approximation runs for ~10 hours, while full HMC takes ~20 hours with $\delta_a = 0.8$ and ~50 hours with $\delta_a = 0.99$, making it difficult to tune the sampler and run our computer experiment.

For computational convenience, we fit the SKIM using only 200 covariates, indexed 2500 - 2700 to encompass the 2586$^{\text{th}}$ covariate which we found to be strongly explanatory. This allows us to easily tune full HMC without altering the takeaways of the experiment. Note that the data here used is different from the data we used in the previous section (since we only examine a subset of the covariates) and the marginal posteriors should therefore not be compared directly.

As in the previous section, we generate 12,000 posterior draws for each method. For full HMC we obtain 36 divergent transitions with $\delta_a = 0.8$ and 0 with $\delta_a = 0.99$. The embedded Laplace approximation produces 0 divergences with $\delta_a = 0.8$. Table 2 shows the covariates which are softly selected. As before, we see a good overlap between full HMC and the embedded Laplace approximation, and mostly disagreeing results from ADVI. Figure 5 compares (i) the posterior draws

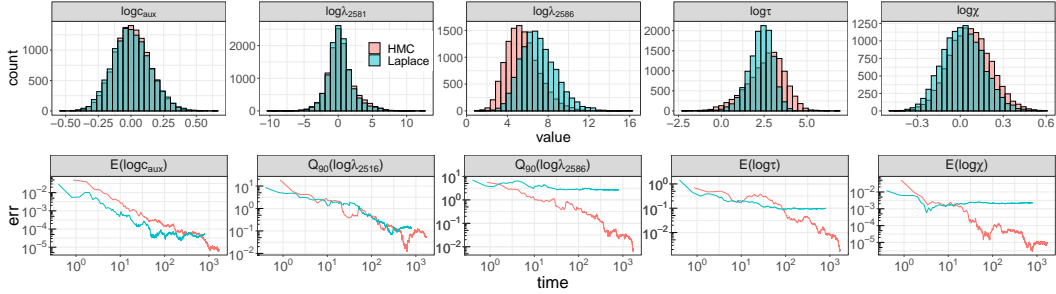

Figure 5: (Up) Samples obtained with full HMC and HMC using an embedded Laplace approximation when fitting the SKIM. (Down) Error when estimating various quantities of interest against wall time. $E$ stands for "expectation" and $Q_{90}$, "90th quantile". Unreported in the figure is that we had to run full HMC twice before obtaining reasonable tuning parameters.

of full HMC and the embedded Laplace approximation, and (ii) the error over time, benchmarked against estimates from 98,000 MCMC iterations, for certain quantities of interest. We obtain comparable estimates but note that the Laplace approximation introduces a bias, which becomes more evident over longer runtimes.

## 7 Discussion

Equipped with a scalable and flexible differentiation algorithm, we expand the regime of models to which we can apply the embedded Laplace approximation. HMC allows us to perform inference even when $\phi$ is high dimensional and multimodal, provided the energy barrier is not too strong. In the case where $\dim(\theta) \gg \dim(\phi)$, the approximation also yields a dramatic speedup. When $\dim(\theta) \ll \dim(\phi)$, marginalizing $\theta$ out can still improve the geometry of the posterior, saving the user time otherwise spent tuning the sampling algorithm. However, when the posterior is well-behaved, the approximation may not provide any benefit.

Our next step is to further develop the prototype for Stan. We are also aiming to incorporate features that allow for a high performance implementation, as seen in the packages INLA, TMB, and GPstuff. Examples include support for sparse matrices required to fit latent Markov random fields, parallelization and GPU support.

We also want to improve the flexibility of the method by allowing users to specify their own likelihood. TMB provides this flexibility but in our view two important challenges persist. Recall that unlike full HMC, which only requires first-order derivatives, the embedded Laplace approximation requires the third-order derivative of the likelihood (but not of the other components in the model). It is in principle possible to apply automatic differentiation to evaluate higher-order derivatives and most libraries, including Stan, support this; but, along with feasibility, there is a question of efficiency and practicality (e.g. [6]): the automated evaluation of higher-order derivatives is often prohibitively expensive. The added flexibility also burdens us with more robustly diagnosing errors induced by the approximation. There is extensive literature on log-concave likelihoods but less so for general likelihoods. Future work will investigate diagnostics such as importance sampling [42], leave-one-out cross-validation [41], and simulation based calibration [37].

## Broader Impact

Through its multidisciplinary nature, the here presented research can act as a bridge between various communities of statistics and machine learning. We hope practitioners of MCMC will consider the benefits of approximate distributions and vice-versa. This work may be a stepping stone to a broader conversation on how, what we have called the two broad approaches of Bayesian computation, can be combined. The paper also raises awareness about existing technologies and may dispel certain misconceptions. For example, our use of the adjoint principle shows that automatic differentiation is not a simple application of the chain rule, but quite a bit more clever than that.

Our goal is to make the method readily available to practitioners across multiple fields, which is why our C++ code and prototype Stan interface are open-source. While there is literature on the Laplace approximation, the error it introduces, and the settings in which it works best, we realize not all potential users will be familiar with it. To limit misuse, we must complement our work with pedagogical material built on the existing references, as well as support and develop more diagnostic tools.

## Acknowledgment

We thank Michael Betancourt, Steve Bronder, Alejandro Catalina, Rok Češnovar, Hyunji Moon, Sam Power, Sean Talts and Yuling Yao for helpful discussions.

CM thanks the Office of Naval Research, the National Science Foundation, the Institute for Education Sciences, and the Sloan Foundation. CM and AV thank the Academy of Finland (grants 298742 and 313122). DS thanks the Canada Research Chairs program and the Natural Sciences and Engineering Research Council of Canada. RA's research was supported in part by a grant from DARPA.

We acknowledge computing resources from Columbia University's Shared Research Computing Facility project, which is supported by NIH Research Facility Improvement Grant 1G20RR030893-01, and associated funds from the New York State Empire State Development, Division of Science Technology and Innovation (NYSTAR) Contract C090171, both awarded April 15, 2010.

We also acknowledge the computational resources provided by the Aalto Science-IT project.

The authors declare to have no conflict of interest.

## Footnotes

[1]More likelihoods can be implemented through a C++ class that specifies the first three derivatives of the log-likelihood.

[2]We do our preliminary runs using only 4000 sampling iterations. The above number are estimated for 12000 sampling iterations. The same holds for the estimated run times.

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
