[Supplementary Material]

# Supplement to "Hamiltonian Monte Carlo using an adjoint-differentiated Laplace approximation: Bayesian inference for latent Gaussian models and beyond"

**Charles C. Margossian**
Department of Statistics
Columbia University
New York, NY 10027
`charles.margossian@columbia.edu`

**Aki Vehtari**
Department of Computer Science
Aalto University
02150 Espoo, Finland
Finnish Center for Artificial Intelligence

**Daniel Simpson**
Department of Statistical Sciences
University of Toronto
ON M5S, Canada

**Raj Agrawal**
CSAIL
Massachusetts Institute of Technology
Cambridge, MA 02139

We review the Newton solver proposed by Rasmussen and Williams [14] and prove theorem 1, the main result required to do build an adjoint method for the embedded Laplace approximation. We next present our prototype code and provide details for the models used in our computer experiments.

## A  Newton solver for the embedded Laplace approximation

---
**Algorithm A** *Newton solver for the embedded Laplace approximation* [14, chapter 3]

---
    **input:** $K$, $y$, $\pi(y \mid \theta, \phi)$
2: $\theta^* = \theta_0$     (initialization)
    **repeat**
4:     $W = -\nabla_\theta \nabla_\theta \log \pi(y \mid \theta^*, \phi)$
        $L = \text{Cholesky}(I + W^{\frac{1}{2}} K W^{\frac{1}{2}})$
6:     $b = W\theta^* + \nabla_\theta \log \pi(y \mid \theta^*, \phi)$
        $a = b - W^{\frac{1}{2}} L^T \setminus (L \setminus (W^{\frac{1}{2}} Kb))$
8:     $\theta^* = Ka$
    **until** convergence
10: $\log \pi(y \mid \phi) = -\frac{1}{2} a^T \theta^* + \log \pi(y \mid \theta^*, \phi) - \sum_i \log L_{ii}$
    **return:** $\theta^*$, $\log \pi_{\mathcal{G}}(y \mid \phi)$

---

Algorithm A is a transcription of the Newton method by Rasmussen and Williams [14, chapter 3] using our notation. As a convergence criterion, we use the change in the objective function between two iterations

$$\Delta \log \pi(\theta \mid y, \phi) \leq \epsilon$$

for a specified $\epsilon$. This is consistent with the approach used in GPStuff [16]. We store the following variables generated during the final Newton step to use them again when computing the gradient: $\theta^*$, $K$, $W^{\frac{1}{2}}$, $L$, and $a$. This avoids redundant computation and spares us an expensive Cholesky decomposition.

## B Building the adjoint method

To compute the gradient of the approximate log marginal with respect to $\phi$, $\nabla \log \pi_{\mathcal{G}}(y \mid \phi)$, we exploit several important principles of automatic differentiation. While widely used in statistics and machine learning, these principles remain arcane to many practitioners and deserve a brief review. We will then construct the adjoint method (theorem 1 and algorithm 2) as a correction to algorithm 1.

### B.1 Automatic differentiation

Given a composite map

$$f = f^L \circ f^{L-1} \circ ... f^1,$$

the chain rule teaches us that the corresponding Jacobian matrix observes a similar decomposition:

$$J = J_L \cdot J_{L-1} \cdot ... \cdot J_1.$$

Based on computer code to calculate $f$, a *forward mode sweep* automatic differentiation numerically evaluates the action of the Jacobian matrix on the initial tangent $u$, or *directional derivative $J \cdot u$*. Extrapolating from the chain rule

$$\begin{aligned} J \cdot u &= J_L \cdot J_{L-1} \cdot ... \cdot J_3 \cdot J_2 \cdot J_1 \cdot u \\ &= J_L \cdot J_{L-1} \cdot ... \cdot J_3 \cdot J_2 \cdot u_1 \\ &= J_L \cdot J_{L-1} \cdot ... \cdot J_3 \cdot u_2 \\ &... \\ &= J_L \cdot u_{L-1}, \end{aligned}$$

where the $u_l$'s verify the recursion relationship

$$\begin{aligned} u_1 &= J_1 \cdot u, \\ u_l &= J_l \cdot u_{l-1}. \end{aligned}$$

If our computation follows the steps outlined above we never need to explicitly compute the full Jacobian matrix, $J_l$, of an intermediate function, $f^l$; rather we only calculate a sequence of Jacobian-tangent products. Similarly a *reverse mode sweep* evaluates the contraction of the Jacobian matrix with a cotangent, $w^T$, yielding $w^T J$, by computing a sequence cotangent-Jacobian products.

Hence, in the case of the embedded Laplace approximation, where

$$\begin{aligned} \mathcal{K}: \quad & \phi & \to K \\ & \mathbb{R}^p & \to \mathbb{R}^{(n+1)n/2} \end{aligned}$$

is an intermediate function, we do not need to explicitly compute $\partial K/\partial \phi$ but only $w^T \partial K/\partial \phi$ for the appropriate cotangent vector. This type of reasoning plays a key role when differentiating functionals of implicit functions – for example, probability densities that depend on solutions to ordinary differential equations – and leads to so-called *adjoint methods* [e.g. 7].

### B.2 Derivation of the adjoint method

In this section we provide a proof of theorem 1. As a starting point, assume algorithm 1 is valid. The proof can be found in Rasmussen and Williams [14, chapter 5]. The key observation is that all operations performed on

$$\frac{\partial K}{\partial \phi_j}$$

are linear. Algorithm 1 produces a map

$$\begin{aligned} \mathcal{Z} \quad & : \partial K/\partial \phi_j & \to \frac{\partial}{\partial \phi_j} \pi(y \mid \phi) \\ & : \mathbb{R}^{n \times n} & \to \mathbb{R}, \end{aligned}$$

and constructs the gradient one element at a time. By linearity,

$$\frac{\partial}{\partial \phi_j} \mathcal{Z}(K) = \mathcal{Z}\left(\frac{\partial K}{\partial \phi_j}\right).$$

**Algorithm B** *Gradient of the approximate marginal log density,* $\log \pi_{\mathcal{G}}(y \mid \phi)$, *with respect to the hyperparameters,* $\phi$, *using reverse mode automatic differentiation*

---

**input:** $y$, $\phi$, $\pi(y \mid \theta, \phi)$
2: Do lines 2 - 6 of Algorithm 2.
  Initiate an expression tree for automatic differentiation with $\phi_v = \phi$.
4: $K_v = \mathcal{K}(\phi_v)$
  $z = \mathcal{Z}(K_v)$
6: Do a reverse-sweep over $z$ to obtain $\nabla_\phi \log \pi(y \mid \phi)$.
  **return:** $\nabla_\phi \log \pi(y \mid \phi)$.

---

Thus an alternative approach to compute the gradient is to calculate the scalar $\mathcal{Z}(K)$ and then use a single reverse mode sweep of automatic differentiation, noting that $\mathcal{Z}$ is an analytical function. This produces Algorithm B. At this point, the most important is done in order to achieve scalability: we no longer explicitly compute $\partial K / \partial \phi$ and are using a single reverse mode sweep.

Automatic differentiation, for all its relatively cheap cost, still incurs some overhead cost. Hence, where possible, we still want to use analytical results to compute derivatives. In particular, we can analytically work out the cotangent

$$w^T := \frac{\partial z}{\partial K}.$$

For the following calculations, we use a lower case, $k_{ij}$ and $r_{ij}$, to denote the $(ij)^{\text{th}}$ element respectively of the matrices $K$ and $R$.

Consider

$$\mathcal{Z}(K) = s_1 + s_2^T s_3,$$

where, unlike in Algorithm 1, $s_1$ and $s_3$ are now computed using $K$, not $\partial K / \partial \phi_j$. We have

$$s_1 = \frac{1}{2} a^T K a - \frac{1}{2} \text{tr}(RK).$$

Then

$$\frac{\partial}{\partial k_{i'j'}} a^T K a = \frac{\partial}{\partial k_{i'j'}} \sum_i \sum_j a_i k_{ij} a_j = a_{i'} a_{j'},$$

and

$$\frac{\partial}{\partial k_{i'j'}} \text{tr}(RK) = \frac{\partial}{\partial k_{i'j'}} \sum_l r_{il} k_{li} = r_{j'i'}.$$

Thus

$$\frac{\partial s_1}{\partial K} = \frac{1}{2} a a^T - \frac{1}{2} R^T.$$

For convenience, denote $l = \nabla_\theta \log \pi(y \mid \theta, \phi)$. We then have

$$b = Kl,$$

$$s_3 = b - \tilde{K} R b = (I - \tilde{K} R) b,$$

where $\tilde{K} = K$, but is maintained fixed, meaning we do not propagate derivatives through it. Let $\tilde{A} = I - \tilde{K} R$ and let $\tilde{a}_{ij}$ denote the $(i, j)^{\text{th}}$ element of $\tilde{A}$. Then

$$s_2^T s_3 = \sum_i (s_2)_i \left( \sum_j \tilde{a}_{ij} \sum_m k_{jm} l_m \right).$$

Thus

$$\frac{\partial}{\partial k_{i'j'}} s_2^T s_3 = \sum_i (s_2)_i \tilde{a}_{ii'} l_{j'} = l_{j'} \sum_i (s_2)_i \tilde{a}_{ii'},$$

where the sum term is the $(i')^{\text{th}}$ element of $\tilde{A}s_2$. The above expression then becomes

$$\frac{\partial}{\partial K}s_2^T s_3 = \tilde{A}s_2 l^T = s_2 l^T - KRs_2 l^T.$$

Combining the derivative for $s_1$ and $s_2^T s_3$ we obtain

$$w^T = \frac{1}{2}aa^T - \frac{1}{2}R + (s_2 + RKs_2)[\nabla_\theta \log \pi(y \mid \theta, \phi)]^T,$$

as prescribed by Theorem 1. This result is general, in the sense that it applies to any covariance matrix, $K$, and likelihood, $\pi(y \mid \theta, \phi)$. Our preliminary experiments, on the SKIM, found that incorporating the analytical cotangent, $w^T$, approximately doubles the differentiation speed.

## C   Computer code

The code used in this work is open source and detailed in this section.

### C.1   Prototype Stan code

The Stan language allows users to specify the joint log density of their model. This is done by incrementing the variable `target`. We add a suite of functions, which return the approximate log marginal density, $\log \pi_\mathcal{G}(y \mid \phi)$. Hence, the user can specify the log joint distribution by incrementing `target` with $\log \pi_\mathcal{G}(y \mid \phi)$ and the prior $\log \pi(\phi)$. A call to the approximate marginal density looks as follows:

```
target +=
  laplace_marginal_*_lpmf (y | n, K, phi, x, delta,
                           delta_int, theta0);
```

The `*` specifies the obervation model, typically a distribution and a link function, for example `bernoulli_logit` or `poisson_log`. The suffix `lpmf` is used in Stan to denote a log posterior mass function. y and n are sufficient statistics for the latent Gaussian variable, $\theta$; K is a function that takes in arguments phi, x, delta, and delta_int and returns the covariance matrix; and theta0 is the initial guess for the Newton solver, which seeks the mode of $\pi(\theta \mid \phi, y)$. Moreover, we have

- y: a vector containing the sum of counts/successes for each element of $\theta$,
- n: a vector with the number of observation for each element of $\theta$,
- K: a function defined in the functions block, with the signature (`vector, data matrix, data real[], data int[]) ==> matrix`. Note that only the first argument may be used to pass variables which depend on model parameters, and through which we propagate derivatives. The term data means an argument may not depend on model parameters.
- phi: the vector of hyperparameters,
- x: a matrix of data. For Gaussian processes, this is the coordinates, and for the general linear regression, the design matrix,
- delta: additional real data,
- delta_int: additional integer data,
- theta0: a vector of initial guess for the Newton solver.

It is also possible to specify the tolerance of the Newton solver. This structure is consistent with other higher-order functions in Stan, such as the algebraic solver and the ordinary differential equation solvers. It gives users flexibility when specifying $K$, but we recognize it is cumbersome. One item on our to-do list is to use variadic arguments, which remove the constraints on the signature of K, and allows users to pass any combination of arguments to K through `laplace_marignal_*_lpmf`.

For each observation model, we implement a corresponding random number generating function, with a call

```
theta =  laplace_marginal_*_rng (y, n, K, phi, x, delta,
                                 delta_int, theta0);
```

This generates a random sample from $\pi_{\mathcal{G}}(\theta \mid y, \phi)$. This function can be used in the generated quantities blocks and is called only once per iteration – in contrast with the target function which is called and differentiated once per integration step of HMC. Moreover the cost of generating $\theta$ is negligible next to the cost evaluating and differentiating $\log \pi(y \mid \phi)$ multiple times per iteration.

The interested reader may find a notebook with demo code, including R scripts and Stan files, at https://github.com/charlesm93/StanCon2020, as part of the 2020 Stan Conference [11].

## C.2   C++ code

We incorporate the Laplace suite of functions inside the Stan-math library, a C++ library for automatic differentiation [4]. The library is open source and available on GitHub, https://github.com/stan-dev/math. Our most recent prototype exists on the branch try-laplace_approximation2[1]. The code is structured around a main function

```
laplace_approximation (likelihood, K_functor, phi, x, delta,
                       delta_int, theta0);
```

with

- likelihood: a class constructed using y and n, which returns the log density, as well as its first, second, and third order derivatives.
- K_functor: a functor that computes the covariance matrix, $K$
- ...: the remaining arguments are as previously described.

A user can specify a new likelihood by creating the corresponding class, meaning the C++ code is expandable.

To expose the code to the Stan language, we use Stan's new OCaml transpiler, stanc3, https://github.com/stan-dev/stanc3 and again the branch try-laplace_approximation2.

Important note: the code is prototypical and currently not merged into Stan's release or development branch.

## C.3   Code for the computer experiment

The code is available on the GitHub public repository, https://github.com/charlems93/laplace_manuscript.

We make use of two new prototype packages: CmdStanR (https://mc-stan.org/cmdstanr/) and posterior (https://github.com/jgabry/posterior).

# D   Tuning dynamic Hamiltonian Monte Carlo

In this article, we use the dynamic Hamiltonian Monte Carlo sampler described by Betancourt [2] and implemented in Stan. This algorithm builds on the No-U Turn Sampler by Hoffman and Gelman [8], which adaptively tunes the sampler during a warmup phase. Hence for most problems, the user does not need to worry about tuning parameters. However, the models presented in this article are challenging and the sampler requires careful tuning, if we do not use the embedded Laplace approximation.

The main parameter we tweak is the *target acceptance rate*, $\delta_a$. To run HMC, we need to numerically compute physical trajectories across the parameter space by solving the system of differential

equations prescribed by Hamilton's equations of motion. We do this using a numerical integrator. A small step size, $\delta$, makes the integrator more precise but generates smaller trajectories, which leads to a less efficient exploration of the parameter space. When we introduce too much numerical error, the proposed trajectory is rejected. Adapt delta, $\delta_a \in (0, 1)$, sets the target acceptance rate of proposed trajectories. During the warmup, the sampler adjusts $\delta$ to meet this target. For well-behaved problems, the optimal value of $\delta_a$ is 0.8 [3].

It should be noted that the algorithm does not necessarily achieve the target set by $\delta_a$ during the warmup. One approach to remedy this issue is to extend the warmup phase; specifically the final fast adaptation interval or *term buffer* [see 8, 15]. By default, the term buffer runs for 50 iterations (when running a warmup for 1,000 iterations). Still, making the term buffer longer does not guarantee the sampler attains the target $\delta_a$. There exist other ways of tuning the algorithm, but at this points, the technical burden on the user is already significant. What is more, probing how well the tuning parameters work usually requires running the model for many iterations.

## E  Automatic differentiation variational inference

ADVI automatically derives a variational inference algorithm, based on a user specified log joint density. Hence we can use the same Stan file we used for full HMC and, with the appropriate call, run ADVI instead of MCMC. The idea behind ADVI is to approximate the posterior over the unconstrained space using a Gaussian distribution, either with a diagonal covariance matrix – leading to a mean-field approximation – or with a full rank covariance matrix. The details of this procedure are described in [10]. Compared to full HMC, ADVI can be much faster, but in general it is difficult to assess how well the variational approximation describes the target posterior distribution without using an expensive benchmark [17, 9]. Furthermore, it can be challenging to assess the convergence of ADVI [6].

To run ADVI, we use the Stan file with which we ran full HMC. We depart from the default tuning parameters by decreasing the learning rate $\eta$ to 0.1, adjusting the tolerance, `rel_tol_obj`, and increasing the maximum number of iterations to 100,000. Our goal is to improve the accuracy of the optimizer as much as possible, while insuring that convergence is reached.

We compare the samples drawn from the variational approximation to samples drawn from full HMC in Figures A, B and C. For the studied examples, we find the approximation to be not very satisfactory, either because it underestimates the posterior variance, does not capture the skewness of the posterior distribution, or returns a unimodal approximation when in fact the posterior density is multimodal. These are all features which cannot be captured by a Gaussian over the unconstrained scale. Naturally, a different choice for $\mathcal{Q}$ could lead to better inference. Using a custom VI algorithm is however challenging, as we need to derive a useful variational family and hand-code the inference algorithm, rather than rely on the implementation in a probabilistic programming language.

## F  Model details

We review the models used in our computer experiments and point the readers to the relevant references.

### F.1  Disease map

The disease map uses a Gaussian process with an exponentiated squared kernel,

$$k(x_i, x_j) = \alpha^2 \exp\left(-\frac{(x_i - x_j)^T (x_i - x_j)}{\rho^2}\right).$$

The full latent Gaussian model is

$$
\begin{aligned}
\rho &\sim \text{invGamma}(a_\rho, b_\rho), \\
\alpha &\sim \text{invGamma}(a_\alpha, b_\alpha), \\
\theta &\sim \text{Normal}(0, K(\alpha, \rho, x)), \\
y_i &\sim \text{Poisson}(y_e^i e^{\theta_i}),
\end{aligned}
$$

Figure A: Samples obtained with full HMC and sampling from the variational approximation produced by ADVI when fitting the disease map. Unlike the embedded Laplace approximation, ADVI strongly disagrees with full HMC.

where we put an inverse-Gamma prior on $\rho$ and $\alpha$.

When using full HMC, we construct a Markov chain over the joint parameter space $(\alpha, \rho, \theta)$. To avoid Neal's infamous funnel [12] and improve the geometry of the posterior distribution, it is possible to use a *non-centered parameterization*:

$$
\begin{aligned}
(\rho, \alpha) &\sim \pi(\rho, \alpha), \\
z &\sim \text{Normal}(0, I_{n \times n}), \\
L &= \text{Cholesky decompose}(K), \\
\theta &= Lz, \\
y_i &\sim \text{Poisson}(y_e^i e^{\theta_i}).
\end{aligned}
$$

The Markov chain now explores the joint space of $(\alpha, \rho, z)$ and the $\theta$'s are generated by transforming the $z$'s. With the embedded Laplace approximation, the Markov chain only explores the joint space $(\alpha, \rho)$.

To run ADVI, we use the same Stan file as for full HMC and set `tol_rel_obj` to 0.005.

## F.2 Regularized horseshoe prior

The horseshoe prior [5] is a sparsity inducing prior that introduces a global shrinkage parameter, $\tau$, and a local shrinkage parameter, $\lambda_i$ for each covariate slope, $\beta_i$. This prior operates a soft variable selection, effectively favoring $\beta_i \approx 0$ or $\beta_i \approx \hat{\beta}_i$, where $\hat{\beta}_i$ is the maximum likelihood estimator. Piironen and Vehtari [13] add another prior to regularize unshrunk $\beta$s, Normal$(0, c^2)$, effectively operating a "soft-truncation" of the extreme tails.

### F.2.1 Details on the prior

For computational stability, the model is parameterized using $c_{\text{aux}}$, rather than $c$, where

$$
c = s_{\text{slab}} \sqrt{c_{\text{aux}}}
$$

with $s_{\text{slab}}$ the slab scale. The hyperparameter is $\phi = (\tau, c_{\text{aux}}, \lambda)$ and the prior

$$
\begin{aligned}
\lambda_i &\sim \text{Student}_t(\nu_{\text{local}}, 0, 1), \\
\tau &\sim \text{Student}_t(\nu_{\text{global}}, 0, s_{\text{global}}), \\
c_{\text{aux}} &\sim \text{inv}\Gamma(s_{\text{df}}/2, s_{\text{df}}/2), \\
\beta_0 &\sim \text{Normal}(0, c_0^2).
\end{aligned}
$$

The prior on $\lambda$ independently applies to each element, $\lambda_i$.

Following the recommendation by Piironen and Vehtari [13], we set the variables of the priors as follows. Let $p$ be the number of covariates and $n$ the number of observations. Additionally, let $p_0$ be the expected number of relevant covariates – note this number does not strictly enforce the number of unregularized $\beta$s, because the priors have heavy enough tails that we can depart from $p_0$. For the

prostate data, we set $p_0 = 5$. Then

$$
\begin{aligned}
s_{\text{global}} &= \frac{p_0}{\sqrt{n}(p - p_0)}, \\
\nu_{\text{local}} &= 1, \\
\nu_{\text{global}} &= 1, \\
s_{\text{slab}} &= 2, \\
s_{\text{df}} &= 100, \\
c_0 &= 5.
\end{aligned}
$$

Next we construct the prior on $\beta$,

$$
\beta_i \sim \text{Normal}(0, \tau^2 \tilde{\lambda}_i^2),
$$

where

$$
\tilde{\lambda}_i^2 = \frac{c^2 \lambda_i^2}{c^2 + \tau^2 \lambda_i^2}.
$$

### F.2.2 Formulations of the data generating process

The data generating process is

$$
\begin{aligned}
\phi &\sim \pi(\phi), \\
\beta_0 &\sim \text{Normal}(0, c_0^2), \\
\beta &\sim \text{Normal}(0, \Sigma(\phi)), \\
y &\sim \text{Bernoulli\_logit}(\beta_0 + X\beta),
\end{aligned}
$$

or, equivalently,

$$
\begin{aligned}
\phi &\sim \pi(\phi), \\
\theta &\sim \text{Normal}(0, c_0^2 I_{n \times n} + X\Sigma(\phi)X^T), \\
y &\sim \text{Bernoulli\_logit}(\theta).
\end{aligned}
$$

For full HMC, we use a non-centered parameterization of the first formulation, much like we did for the disease map. The embedded Laplace approximation, as currently implemented, requires the second formulation, which is mathematically more convenient but comes at the cost of evaluating and differentiating $K = c^2 I_{n \times n} + X\Sigma(\phi)X^T$. In this scenario, the main benefit of the Laplace approximation is not an immediate speed-up but an improved posterior geometry, due to marginalizing $\theta$ (and thus implicitly $\beta$ and $\beta_0$) out. This means we do not need to fine tune the sampler.

### F.2.3 Fitting the model with full HMC

This section describes how to tune full HMC to fit the model at hand. Some of the details may be cumbersome to the reader. But the takeaway is simple: tuning the algorithm is hard and can be a real burden for the modeler.

Using a non-centered parameterization and with Stan's default parameters, we obtain ~150 divergent transitions[2]. We increase the target acceptance rate to $\delta_a = 0.99$ but find the sampler now produces 186 divergent transitions. A closer inspection reveals the divergences all come from a single chain, which also has a larger adapted step size, $\delta$. The problematic chain also fails to achieve the target acceptance rate. These results are shown in Table 1. From this, it seems increasing $\delta_a$ yet again may not provide any benefits. Instead we increase the term buffer from 50 iterations to 350 iterations. With this setup, we however obtain divergent transitions across all chains.

This outcome indicates the chains are relatively unstable and emphasizes how difficult it is, for this type of model and data, to come up with the right tuning parameters. With $\delta_a = 0.999$ and the extended term buffer we observe 13 divergent transitions. It is possible this result is the product of luck, rather than better tuning parameters. To be clear, we do not claim we found the optimal model parameterization and tuning parameters. There is however, to our knowledge, no straightforward way to do so.

Table 1: Adapted tuning parameters across 4 Markov chains with $\delta_a = 0.99$.

| Chain | Step size | Acceptance rate | Divergences |
|---|---|---|---|
| 1 | 0.0065 | 0.99 | 0 |
| 2 | 0.0084 | 0.90 | 186 |
| 3 | 0.0052 | 0.99 | 0 |
| 4 | 0.0061 | 0.99 | 0 |

Figure B: Samples obtained with full HMC and sampling from the variational approximation produced by ADVI when fitting a general linear model with a regularized horseshoe prior.

### F.2.4 Fitting the model with the embedded Laplace approximation

Running the algorithm with Stan's default tuning parameters produces 0 divergent transitions over 12,000 sampling iterations.

### F.2.5 Fitting the model with ADVI

To run ADVI, we use the same Stan file as for full HMC and set `tol_rel_obj` to 0.005.

The family of distribution, $\mathcal{Q}$, over which ADVI optimizes requires the exact posterior distribution to be unimodal over the unconstrained scale. This is a crucial limitation in the studied example, as shown in Figure B. This notably affects our ability to select relevant covariates using the $90^{\text{th}}$ posterior quantile. When examining the top six selected covariates (Table 1 in the main text), we find the result from ADVI to be in disagreement with full HMC and the embedded Laplace approximation. In particular, $\lambda_{2586}$ which corresponds, according to our other inference methods, to the most relevant covariate, has a relatively low $90^{\text{th}}$ quantile. This is because ADVI only approximates the smaller mode of $\pi(\lambda_{2586} \mid y)$. Our results are consistent with the work by Yao et al. [17], who examine ADVI on a similar problem.

### F.3 Sparse kernel interaction model

SKIM, developed by Agrawal et al. [1], extends the model of Piironen and Vehtari [13] by accounting for pairwise interaction effects between covariates. The generative model shown below uses the notation in F.2 instead of that in Appendix D of Agrawal et al. [1]:

$$\chi \sim \text{inv}\Gamma(s_{\text{df}}/2, s_{\text{df}}/2),$$
$$\eta_2 = \frac{\tau^2}{c^2}\chi,$$
$$\beta_i \mid \tau, \tilde{\lambda} \sim \text{Normal}(0, \tau^2 \tilde{\lambda}_i^2),$$
$$\beta_j \mid \tau, \tilde{\lambda} \sim \text{Normal}(0, \tau^2 \tilde{\lambda}_i^2),$$
$$\beta_{ij} \mid \eta_2, \tilde{\lambda} \sim \text{Normal}(0, \eta_2^2 \tilde{\lambda}_i^2 \tilde{\lambda}_j^2),$$
$$\beta_0 \mid c_0^2 \sim \text{Normal}(0, c_0^2),$$

where $\beta_i$ and $\beta_{ij}$ are the main and pairwise effects for covariates $x_i$ and $x_i x_j$, respectively, and $\tau, \tilde{\lambda}, c_0$ are defined in F.2.

Figure C: Samples obtained with full HMC and sampling from the variational approximation produced by ADVI when fitting the SKIM.

Instead of sampling $\{\beta_i\}_{i=1}^p$ and $\{\beta_{ij}\}_{i,j=1}^p$, which takes at least $O(p^2)$ time per iteration to store and compute, Agrawal et al. [1] marginalize out all the regression coefficients, only sampling $(\tau, \xi, \tilde{\lambda})$ via MCMC. Through a kernel trick and a Gaussian process re-parameterization of the model, this marginalization takes $O(p)$ time instead of $O(p^2)$. The Gaussian process covariance matrix $K$ induced by SKIM is provided below:

$$
\begin{aligned}
K_1 &= x \operatorname{diag}(\tilde{\lambda}^2)\, x^T, \\
K_2 &= [x \circ x]\operatorname{diag}(\tilde{\lambda}^2)\, [x \circ x]^T,
\end{aligned}
$$

where "$\circ$" denotes the element-wise Hadamard product. Finally,

$$
\begin{aligned}
K &= \frac{1}{2}\eta_2^2(K_1+1)\circ(K_1+1) - \frac{1}{2}\eta_2^2 K_2 - (\tau^2 - \eta_2^2)K_1 \\
&\quad + c_0^2 - \frac{1}{2}\eta_2^2.
\end{aligned}
$$

## Footnotes

[1]Our first prototype is was on the branch try-laplace_approximation, and was used to conduct the here presented computer experiment. The new branch modifies the functions' signatures to be more consistent with the Stan language. In this Supplement, we present the new signatures.

[2]To be precise, we here did a preliminary run using 4000 sampling iterations and obtained 50 divergent transitions (so an expected 150 over 12000 sampling iterations).