[Reviews · NeurIPS 2020]

Review 1

Summary and Contributions: This paper introduces an efficient and fast approach to marginalising out the latent variables in Gaussian latent variable models by using an embedded Laplace approximation. The paper then provides an adjoint method to enable derivatives of the approximate log marginal density to be used in HMC for inferring the approximate posterior over the hyperparameters. ############### Response to Author Feedback ############### Thanks very much for the author response and directly responding to my questions. It was very helpful to hear your comments on the Hessian. I agree with the authors that actually implementing an efficient (fair) version of work such as semi-separable HMC would be a challenge and also I agree that it has not seen much wide use. I also understand that including a way of representing tuning time is challenging, but if you can think of one, I think it would help (even mentioning the number of times to fit the model would definitely be useful). Just a comment: to overcome the issue of a GP having few hyperparameters, it might be possible to work with kernels that have a much larger number of hyperparameters, such as spectral kernels e.g "Gaussian Process Kernels for Pattern Discovery and Extrapolation" by Wilson and Adams (2013).

Strengths: The paper is written well and its relation to previous work is clear. The introduction is well-motivated and the combination of the Laplace approximation from Rasmussen and Williams with the use of the adjoint method is described in a way that makes the story of the paper easy to follow.

Weaknesses: My main questions regarding the paper: 1) When computing the Laplace approximation, this still requires calculation of the Hessian, which I believe is with respect to the latent (theta). This is referred to as W in Algorithm 1. Would it be possible to comment further on the kind of trade-off between implementing full-HMC, versus the overhead of calculating the Hessian. I think this is the issue you are referring to in the second paragraph of the discussion section, whereby you mention higher-order automatic differentiation. (I.e. I assume you stick to analytical Hessians (e.g. Table on Page 43 Rasmussen and Williams)). 2) Are there any alternative methods available that you have explored that marginalise out the latent variables? For example “Semi-Separable Hamiltonian Monte Carlo for Inference in Bayesian Hierarchical Models” by Zhang and Sutton jointly sample over hyperparameters and parameters to overcome similar funnel-like behaviours to that of the Gaussian latent variable models that you explore. Although not strictly necessary in this work, it might be interesting to compare. 3) While Figure 1 and the GP with a Poisson likelihood both successfully show the speed of the new approach, the results of Sections 5 and 6 are slightly harder to interpret in terms of the performance benefits. Both Figures show that full HMC’s error often drops in less time than the embedded Laplace method. I think it would be helpful to put in the overhead time in tuning full HMC somewhere in the plots to really highlight how much of a headache tuning is! 4) When the Laplace approximation is introduced (approx. Line 62), is there any implicit approximation that the joint posterior can be written as p(phi | y) p(theta | phi, y)? Is this an assumption being made and does this result in any limitations? (I may be wrong and it may be possible to decompose the posterior in this way for a Gaussian latent variable model but I would be interested to hear from the authors.)

Correctness: The paper seems correct.

Clarity: The paper is very well written and clear.

Relation to Prior Work: The relation to prior work is clear.

Reproducibility: Yes

Additional Feedback: The authors provide an extensive supplementary material and they also provide the code.


Review 2

Summary and Contributions: Performing Bayesian inference on Gaussian latent variable models can be challenging since MCMC algorithms struggle with the geometry of the posterior and can be prohibitively slow. One potential solution is to use a Laplace approximation to marginalize out the latent Gaussian variables and then integrate out the remaining hyperparameters using HMC. To implement this method efficiently, the authors derive a novel adjoint method that propagates the minimal information needed to construct the gradient of the approximate marginal likelihood

Strengths: 1. They discuss and analysis the disadvantages of the existing HMC methods on Gaussian latent variable models, which provide a clear motivation of their work. 2. Via simple and easy-to-understand theorical derivation, they propose a new method to marginalize using a Laplace approximation.

Weaknesses: 1. What does K means in the beginning of introduction? It makes me confusion. 2. The compared method in experiments is just the standard HMC. Is there any existing methods as a baseline to illustrate the advantages of their proposed method?

Correctness: I am not very familiar with this field. I try my best to understand their method and proofs. I think it is correct.

Clarity: The paper is well written.

Relation to Prior Work: Clearly.

Reproducibility: Yes

Additional Feedback:


Review 3

Summary and Contributions: The submission considers approximate Bayesian inference on latent Gaussian models by marginalizing out the latent Gaussian variables using a Laplace approximation and performing HMC for the hyper-parameters. The new contribution is that the gradient of the approximate log marginal posterior can be computed efficiently with automatic differentiation via vector-Jacobian products instead of previous work that computes the full Jacobian of the covariance function. ################################## POST AUTHOR RESPONSE UPDATE Having read the response from the authors as well as the other reviews, I will keep my weak accept score. A concern raised in my review was a lack of comparison to alternative approximate inference methods such as variational inference approaches, as also raised in review 4. The authors refer in their response to work comparing the embedded Laplace approximation to variational inference from 10 years ago. Arguably, there was some work improving variational inference approaches in the meantime that also have different trade-offs of complexity/performance vs computing time – and an empirical assessment as to where the proposed approach lies in this respect would be very helpful in a revised version. The authors also responded to include more details on the comparison with the adaptive HMC benchmark. ################################################

Strengths: The proposed method allows for approximate Bayesian inference over very high dimensional hyper-parameters in latent Gaussian models. The idea of differentiating through a Newton-solver and using the fact that this solver is linear in the gradient of the covariance function, is new as far as I am aware. Gaussian latent variables (with many hyper-parameters) are used frequently, so this method - as implemented currently in the Stan framework - can be quite relevant to the NeurIPS community.

Weaknesses: The paper evaluates the performance of the proposed method on three models with log-concave targets only, comparing it to HMC/NUTS only. It would be useful to add some experimental evaluation on more general likelihood function, particularly when the Laplace approximation becomes less accurate (targets that are multi-modal, non-constant curvature) and compare it with alternative approximate methods such as variational ones in this case. In the case where marginalizing the latent variable might lead to "nicer" targets, it would be interesting to compare the method to some additional adaptive MCMC methods - it is not so clear to me what adaptation is used on the bench-marked full HMC sampler (I guess the number of leap-frog steps and the step-size, but also some (non-diagonal) mass matrix/pre-conditioning?)

Correctness: The claims and empirical methodology seem correct.

Clarity: The paper is written relatively well.

Relation to Prior Work: Related prior work seems to be discussed properly.

Reproducibility: Yes

Additional Feedback: I was wondering if this gradient calculation is related to work that consider implicit gradients of hyper-parameters where the parameters follow SGD dynamics (Lorraine et al., Optimizing Millions of Hyperparameters by Implicit Differentiation, 2020; Shaban et al., Truncated Back-propagation for Bilevel Optimization, 2019)? Minor remark: What is Z following line 31 (Appendix) exactly, I assume it is different from Z defined in line 42, which seems to be not a linear function (but has a quadratic term)? The notations \pi(y|\phi) and \pi_{\mathcal{G}}(y|\phi) seem to be used quite interchangeably in the text/algorithm - is this intended?


Review 4

Summary and Contributions: The paper proposes a novel inference algorithms for Gaussian Process models with a very large number of hyperparameters. The approach involves using a Laplace approximation of the latent variables and a novel computation for the gradient of the target density w.r.t. the hyperparameters by computing the gradient's inner product with an appropriately chosen adjoint. The paper demonstrates that their approach works faster that pure MCMC even in cases when the number of hyperparameters are small. ====== POST REBUTTAL ============ The lack of comparison to *recent* VI work in GPs and in particular gpytorch remains a major drawback of the paper. Irrespective of the misnomer around ARD the unaddressed concern is that this work is not well motivated. Recasting an ARD problem to a GP with a diagonal covariance just so we can get an example with a large number of hyperparameters is not a great motivation for work on GPs IMHO.

Strengths: The claims in the paper appear sound. The calculation of the gradient of the target density w.r.t. the hyperparameters is essentially a rearrangement of terms in the classical literature to allow for a single pass of reverse-mode autodifferentiation. This seems like a very simple contribution, but it adds to our knowledge of GPs. The experiments demonstrate clearly that the described optimization helps in better runtime performance and for large number of hyperparameters the improvements are considerably better than pure MCMC. The work is somewhat significant and novel since it fills a missing hole in Laplace approximated Gaussian Process research when the number of hyperparameters are very large. The class of regression models where we are trying to determine the relevance of covariates is relevant to the NeurIPS community. Also a lot of work in the NeurIPS community has recently focused on Gaussian Processes.

Weaknesses: Since this paper falls in the class of approximate inference for Gaussian Processes it would have been appropriate to compare it to Variational Inference. In particular the gpytorch package which is frequently cited in GP research would have been appropriate to compare against. So it is unclear whether the improvements here as compared to pure MCMC would stack up better when compared to other approximate inference techniques. As far as signficance, it is very unusual for Gaussian Processes to have such a large number of hyperparameters. It is not well motivated as to why this is an interesting problem to solve. For example the problem in section 5 where a local scale term is introduced for each of the 6000 covariates seems like a typical problem in the Automatic Relevance Determination (ARD) class of models. ARD was also not referenced or compared against. It is not clear why we should recast this problem as a GP with a diagonal covariance?

Correctness: The claims and methods appear correct.

Clarity: The paper is very clearly written.

Relation to Prior Work: Comparison to VI or ARD was not provided, so the significance of the work is unclear.

Reproducibility: Yes

Additional Feedback: Line 18 describes the model structure as "latent Gaussian model:. But there is no agreement in the literature to justify this name. For example Deep Gaussian models assume that a sample from a standard normal is the first sampled variable followed by a deep net. While the model in line 18 uses a multivariate normal for the latent variable which is the second sampled variable. To be honest, the structure of this model looks exactly like a Gaussian Process. For example gpytorch uses exactly this structure. Alternatively, please provide a citation to defend the name Latent Gaussian Model. On Line 50, \pi is used for both the posterior and the prior. My suggestion would be to use `p` for the specifying the prior and `\pi` for the unnormalized posterior which is the more familiar convention to this reviewer. On Line 67 it is probably worth clarifying that \theta* is the mode "conditional on \phi". Line 99, although K could be arbitrary, in practice the possible choices of K for GPs come from a very small set, so its not clear how impactful this work is.

[Author Response · NeurIPS 2020]

**Higher-order derivatives of likelihood.** As proposed by Reviewer 1, we will comment that, unlike full HMC, the embedded Laplace approximation (ELA) requires the Hessian w.r.t $\theta$, and furthermore, the third-order derivatives (line 6 in algorithm 1). It is possible to obtain these derivatives using automatic differentiation but only at an important cost, which is why we stick to analytical derivatives for the log likelihood. Our intuition is that we can bypass this calculation but this result is not in the present work. This is why we have not, as noted by Reviewer 3, run computer experiments on general likelihoods, but we agree that this is exactly the right direction for future research.

**Choice of benchmark.** We use adaptive HMC as our benchmark because for our main motivating problems, in Sections 5 and 6, this is the method used in references. The authors of these models choose MCMC in order to accommodate a high-dimensional and intricate prior, built as a mixture of Cauchy distributions. We will, as suggested by Reviewer 3, detail which tuning parameters of HMC get adapted, likely in the Supplementary. As Reviewers 3 and 4's comments imply, the reader will ask: "why use the Laplace approximation and not variational inference?" There already exist several papers that compare the ELA to variational inference in various settings, for example Nickish and Rasmussen [2008], Vanhatalo and Vehtari [2010]. Their analysis finds that marginalization with ELA yields comparable, and at times better performance, for several problems of interest. Section 1.1 will mention variational inference (notably its implementation in gppytorch) and point the reader to the above references. With that said, our paper's focus remains on an improvement to the Laplace approximation and to HMC.

**ARD.** Reviewer 4 correctly indicates automatic relevance determination regression (ARD) is relevant to the problems we study. The regularized horseshoe model and the SKIM are in fact ARD models with a special choice of sparsity inducing prior. We avoided the term ARD because it is a misnomer, as in most used cases it assesses non-linearity and not relevance, see e.g. Paananen et al. [2019]. Nevertheless we will draw the connection to ARD for the benefit of the readers.

**Impact and applications.** Reviewer 4 points out that typically the choice of $K$ comes from a small subset and that the dimension of the hyperparameter is small. We agree, and our method is not meant to supersede existing methods for classic GPs. Our goal is to empower users to fit non-classic, but in our view nevertheless important, GPs, such as those presented in Sections 5 and 6; notably the SKIM which has a non-trivial $K$. Furthermore, many popular hierarchical models can be cast as latent Gaussian models – as done for example in Section 5 and Section E.2 of the Supplement – which makes for a relatively broad range of applications.

**Naming convention.** To justify the name "latent Gaussian model", we will state that we follow the convention in Rue and Chopin [2009] and emphasize the term is used differently across the literature. Per Reviewer 2's suggestion, we will state that $K$ is the covariance matrix.

**Including user time in figures.** We appreciate Reviewer 1's comments on figures 4 and 5. Papers often compare the computational run times of (hopefully) well-tuned algorithms, with less consideration for user time to tune said algorithms. We are not sure how to best measure this latter factor and include it in the figures. Our plan is to mention in the captions how many times *we* had to fit each model to tune it. We will tinker with better alternatives.

**Related work on adaptive HMC.** Based on comments by Reviewers 1 and 3, it seems readers would appreciate a brief discussion on adaptive HMC methods; notably semi-separable HMC by Zhang and Sutton [2014] and RHMC. These methods are indeed designed to overcome the geometric difficulties our paper is concerned with. We did not use them as benchmarks because they are either not commonly used, computationally expensive, or because building an efficient implementation in C++ – to make comparisons equitable – is not straightforward. We will however mention these contributions either in Section 1.1 or in Section D of the supplement.

**Related work on implicit differentiation.** Reviewer 3 refers us to the work by Lorraine et al. [2019], which uses the implicit function theorem (IFT), as one would when differentiating a black box optimizer (notably mentioned in Section 2 of our submission). The IFT requires an inverse Hessian term which must be fully computed and a Hessian term that can be contracted with a co-tangent vector. Because you have to compute the full inverse Hessian, a good approximation is desirable. We considered using IFT but after some experimentation chose to not treat the optimizer as a black box and use an "open" Newton solver instead. This allowed us to bypass the computation of the inverse Hessian altogether. Hence IFT is a valid choice but we have found our approach to work better for our problem. The comparison is interesting and may be worth describing in the Supplement.

**References.** (Nickis and Rasmussen, 2008) *Approximations for binary Gaussian process classification*, JMLR; (Vanhatalo and Vehtari, 2010) *Speeding up the binary Gaussian process classification*, Uncertainty in Artificial Intelligence; (Rue and Chopin, 2009) *Approximate Bayesian inference for latent Gaussian models by using integrated nested Laplace approximations*, Journal of Royal Statistics B; (Lorraine et al, 2020) *Optimizing millions of hyperparameters by implicit differentiation*, Artificial Intelligence and Statistics; (Zhang and Sutton, 2014) *Semi-Separable Hamiltonian Monte Carlo for Inference in Bayesian Hierarchical Models*, Neurips; (Paanen et al, 2019) *Variable selection for Gaussian processes via sensitivity analysis of the posterior predictive distribution*, PMLR.

[Meta-Review · NeurIPS 2020]

This paper develops an inference method for Gaussian latent variable models that employs a Laplace approximation marginalize over latent variables and infer hyperparameters using HMC. The authors use an adjoint method to efficiently compute the gradient with respect to the hyperparameters. The main contribution is that inference can scale to hyperparameters that have a high dimensionality (>1000). This paper was overall well-received by reviewers, who remained on balance in favor of acceptance after the author response. The main outstanding points of criticism, which the AC would like to encourage the authors to address are that: (1) the authors should more clearly motivate the use case for latent Gaussian models with a large number of parameters (2) discussion of recent advances in variational inference for GPs is warranted (and some form of comparison would be appreciated). As a final comment, the AC and reviewers would like to suggest that the authors revise the title of this manuscript to include mention of latent Gaussian models, since the proposed method is specific to these models.